# Weitzman's Rule for Pandora's Box with Correlations

**Evangelia Gergatsouli**
University of Wisconsin-Madison
evagerg@cs.wisc.edu

**Christos Tzamos**
University of Wisconsin-Madison
& University of Athens
tzamos@wisc.edu

## Abstract

PANDORA'S BOX is a central problem in decision making under uncertainty that can model various real life scenarios. In this problem we are given $n$ boxes, each with a fixed opening cost, and an unknown value drawn from a known distribution, only revealed if we pay the opening cost. Our goal is to find a strategy for opening boxes to minimize the sum of the value selected and the opening cost paid.

In this work we revisit PANDORA'S BOX when the value distributions are correlated, first studied in Chawla et al. [2020]. We show that the optimal algorithm for the independent case, given by Weitzman's rule, directly works for the correlated case. In fact, our algorithm results in significantly improved approximation guarantees compared to the previous work, while also being substantially simpler. We also show how to implement the rule given only sample access to the correlated distribution of values. Specifically, we find that a number of samples that is polynomial in the number of boxes is sufficient for the algorithm to work.

## 1  Introduction

In various minimization problems where uncertainty exists in the input, we are allowed to obtain information to remove this uncertainty by paying an extra price. Our goal is to sequentially decide which piece of information to acquire next, in order to minimize the sum of the search cost and the value of the option we chose.

This family of problems is naturally modeled by PANDORA'S BOX, first formulated by Weitzman [1979] in an economics setting, with multiple application in consumer search, housing markets and job search (see [McCall and McCall, 2007] for more applications). In this problem where we are given $n$ boxes, each containing a value drawn from a known distribution and each having a fixed known *opening cost*. We can only see the exact value realized in a box if we open it and pay the opening cost. Our goal is to minimize the sum of the value we select and the opening costs of the boxes we opened.

In the original work of Weitzman, an optimal solution was proposed when the distributions on the values of the boxes were independent [Weitzman, 1979]. This algorithm was based on calculating a *reservation value* $(\sigma)$ for each box, and then choosing the box with the lowest reservation value to open at every step. Independence, however, is an unrealistic assumption in real life; in a housing market neighboring houses' price are affected the same way, or in a job search setting, candidates might share qualifications that affect them similarly. Wanting to tackle a more realistic setting, Chawla et al. [2020] first studied the problem where the distributions are correlated, and designed an algorithm giving a constant approximation guarantee. This algorithm is quite involved, it requires solving an LP to convert the PANDORA'S BOX instance to a MIN SUM SET COVER one, and then solving this instance to obtain an ordering of opening the boxes. Finally, it reduces the problem of deciding when to stop to an online algorithm question corresponding to SKI-RENTAL.

37th Conference on Neural Information Processing Systems (NeurIPS 2023).

## 1.1 Our Contribution

In this work we revisit PANDORA'S BOX with correlations, and provide **simpler**, **learnable** algorithms with **better approximation guarantees**, that directly **generalize** Weitzman's reservation values. More specifically, our results are the following.

- **Generalizing**: we first show how the original reservation values given by Weitzman [1979] can be generalized to work in correlated distributions, thus allowing us to use a version of their initial greedy algorithm.

- **Better approximation**: we give two different variants of our main algorithm, that each uses different updates on the distribution $\mathcal{D}$ after every step.
  1. *Variant 1: partial updates*. We condition on the algorithm not having stopped yet.
  2. *Variant 2: full updates*. We condition on the exact value $v$ revealed in the box opened.

  Both variants improve the approximation given by Chawla et al. [2020] from $9.22$ to $4.428$ for Variant 1 and to $5.828$ for Variant 2. It is worth noting that our result for Variant 1 is *almost tight*, since the best possible approximation factor we can obtain is $4$, implied by Feige [1998]. We include more details on the lower bound in Section A.4 of the Appendix.

- **Simplicity**: our algorithms are greedy and only rely on the generalized version of the reservation value, while the algorithms in previous work rely on solving a linear program, and reducing first to MIN SUM SET COVER then to SKI-RENTAL, making them not straightforward to implement. A $9.22$ approximation was also given in Gergatsouli and Tzamos [2022], which followed the same approach but bypassed the need to reduce to MIN SUM SET COVER by directly rounding the linear program via randomized rounding.

- **Learnability**: we show how given sample access to the correlated distribution $\mathcal{D}$ we are able to still maintain the approximation guarantees. Specifically, for Variant 1 only $\text{poly}(n, 1/\varepsilon, \log(1/\delta))$ samples are enough to obtain $4.428 + \varepsilon$ approximation with probability at least $1 - \delta$. Variant 2 is however impossible to learn.

Our analysis is enabled by drawing similarities from PANDORA'S BOX to MIN SUM SET COVER, which corresponds to the special case of when the values inside the boxes are $0$ or $\infty$. For MIN SUM SET COVER a simple greedy algorithm was shown to achieve the optimal $4$-approximation [Feige et al., 2002]. Surprisingly, Weitzman's algorithm can be seen as a direct generalization of that algorithm. Our analysis follows the histogram method introduced in Feige et al. [2002], for bounding the approximation ratio. However, we significantly generalize it to handle values in the boxes and work with tree-histograms required to handle the case with full-updates.

## 1.2 Related Work

Since Weitzman's initial work [Weitzman, 1979] on PANDORA'S BOX there has been a renewed interest in studying this problem in various settings. Specifically Doval [2018], Beyhaghi and Kleinberg [2019], Beyhaghi and Cai [2023a], Fu et al. [2023] study PANDORA'S BOX when we can select a box without paying for it (non-obligatory inspection), in Boodaghians et al. [2020] there are tree or line constraints on the order in which the boxes can be opened. In Chawla et al. [2020, 2021] the distributions on the values inside the boxes are correlated and the goal is to minimize the search and value cost, while finally in Bechtel et al. [2022] the task of searching over boxes is delegated by an agent to a principal, while the agent makes the final choice. The recent work of Chawla et al. [2020] is the first one that explores the correlated distributions variant and gives the first approximation guarantees. The recent survey by Beyhaghi and Cai [2023b] summarizes the recent work on Pandora's Box and its variants.

This problem can be seen as being part of the "price of information" literature [Charikar et al., 2000, Gupta and Kumar, 2001, Chen et al., 2015b,a], where we can remove part of the uncertainty of the problem at hand by paying a price. In this line of work, more recent papers study the structure of approximately optimal rules for combinatorial problems [Goel et al., 2006, Gupta and Nagarajan, 2013, Adamczyk et al., 2016, Gupta et al., 2016, 2017, Singla, 2018, Gupta et al., 2019].

For the special case of MIN SUM SET COVER, since the original work of Feige et al. [2002], there has been many follow-ups and generalizations where every set has a requirement of how many elements

contained in it we need to choose [Azar et al., 2009, Bansal et al., 2010, Azar and Gamzu, 2011, Skutella and Williamson, 2011, Im et al., 2014].

Note also that multiple results on problems related to Pandora's box have been published in ML-related conferences, as this is a problem that encompasses both algorithmic and learning aspects (e.g. Esfandiari et al. [2019], Gergatsouli and Tzamos [2022], Bhaskara et al. [2020], Cesa-Bianchi et al. [2021], Guo et al. [2021]).

## 2 Preliminaries

In PANDORA'S BOX ($\mathcal{PB}$) we are given a set of $n$ boxes $\mathcal{B}$, each with a known opening cost $c_b \in \mathbb{R}^+$, and a distribution $\mathcal{D}$ over a vector of unknown values $\boldsymbol{v} = (v_1, \ldots, v_n) \in \mathbb{R}^n_+$ inside the boxes. Each box $b \in \mathcal{B}$, once it is opened, reveals the value $v_b$. The algorithm can open boxes sequentially, by paying the opening cost each time, and observe the value instantiated inside the box. The goal of the algorithm is to choose a box of small value, while spending as little cost as possible "opening" boxes. Formally, denoting by $\mathcal{O} \subseteq \mathcal{B}$ the set of opened boxes, we want to minimize

$$\mathbb{E}_{v \sim \mathcal{D}} \left[ \min_{b \in \mathcal{O}} v_b + \sum_{b \in \mathcal{O}} c_b \right].$$

A *strategy* for PANDORA'S BOX is an algorithm that in every step decides which is the next box to open and when to stop. We measure the performance of our algorithm usign the competitive (or approximation) ratio; a strategy $\mathcal{A}$ is $\alpha$-approximation if $\mathbb{E}[\mathcal{A}] \leq \alpha\text{OPT}$, where OPT is the optimal online algorithm[1]

A strategy can pick any open box to select at any time. To model this, we assume without loss of generality that after a box is opened the opening cost becomes $0$, allowing us to select the value without opening it again. In its full generality, a strategy can make decisions based on every box opened and value seen so far. We call this the *Fully-Adaptive* (FA) strategy.

**Different Benchmarks.** As it was initially observed in Chawla et al. [2020], optimizing over the class of fully-adaptive strategies is intractable, therefore we consider the simpler benchmark of *partially-adaptive* (PA) strategies. In this case, the algorithm has to fix the opening order of the boxes, while the stopping rule can arbitrarily depend on the values revealed.

### 2.1 Weitzman's Algorithm

When the distributions of values in the boxes are independent, Weitzman [1979] described a greedy algorithm that is also the optimal strategy. In this algorithm, we first calculate an index for every box $b$, called *reservation value* $\sigma_b$, defined as the value that satisfies the following equation

$$\mathbb{E}_{\boldsymbol{v} \sim \mathcal{D}} \left[ (\sigma_b - v_b)^+ \right] = c_b, \tag{1}$$

where $(a - b)^+ = \max(0, a - b)$. Then, the boxes are ordered by increasing $\sigma_b$ and opened until the minimum value revealed is less than the next box in the order. Observe that this is a *partially-adaptive* strategy.

## 3 Competing with the Partially-Adaptive

We begin by showing how Weitzman's algorithm can be extended to correlated distributions. Our algorithm calculates a reservation value $\sigma$ for every box at each step, and opens the box $b \in \mathcal{B}$ with the minimum $\sigma_b$. We stop if the value is less than the reservation value calculated, and proceed in making this box *free*; we can re-open this for no cost, to obtain the value just realized at any later point. The formal statement is shown in Algorithm 1.

We give two different variants based on the type of update we do after every step on the distribution $\mathcal{D}$. In the case of partial updates, we only condition on $V_b > \sigma_b$, which is equivalent to the algorithm

---

[1]The optimal online has the exact same information as our algorithm $\mathcal{A}$ but has infinite computation time to solve the problem.

not having stopped. On the other hand, for full updates we condition on the exact value that was instantiated in the box opened. Theorem 3.1 gives the approximation guarantees for both versions of this algorithm.

---

**Algorithm 1:** Weitzman's algorithm, for correlated $\mathcal{D}$.

---

**Input:** Boxes with costs $c_i \in \mathbb{R}$, distribution over scenarios $\mathcal{D}$.

1 An unknown vector of values $v \sim \mathcal{D}$ is drawn
2 **repeat**
3     Calculate $\sigma_b$ for each box $b \in \mathcal{B}$ by solving:

$$\mathbb{E}_{\boldsymbol{v} \sim \mathcal{D}} \left[ (\sigma_b - v_b)^+ \right] = c_b.$$

4     Open box $b = \operatorname{argmin}_{b \in \mathcal{B}} \sigma_b$
5     Stop if the value the observed $V_b = v_b \leq \sigma_b$
6     $c_b \leftarrow 0$ // Box is always open now or can be reopened
7     Update the prior distribution
        - **Variant 1**: $\mathcal{D} \leftarrow \mathcal{D}|_{V_b > \sigma_b}$ (partial updates)
        - **Variant 2**: $\mathcal{D} \leftarrow \mathcal{D}|_{V_b = v_b}$ (full updates)

8 **until** *termination*;

---

**Theorem 3.1.** *Algorithm 1 is a $4.428$-approximation for Variant 1 and $5.828$-approximation for Variant 2 of* PANDORA'S BOX *against the partially-adaptive optimal.*

*Proof.* We seperately show the two components of this theorem in Theorems 3.2 and 3.3. $\square$

Observe that for independent distributions this algorithm is exactly the same as Weitzman's [Weitzman, 1979], since the product prior $\mathcal{D}$ remains the same, regardless of the values realized. Therefore, the calculation of the reservation values does not change in every round, and suffices to calculate them only once at the beginning.

**Scenarios**     To proceed with the analysis of Theorem 3.1, we assume that $\mathcal{D}$ is supported on a collection of $m$ vectors, $(\boldsymbol{v}^s)_{s \in \mathcal{S}}$, which we call scenarios, and sometimes abuse notation to say that a scenario is sampled from the distribution $\mathcal{D}$. We assume that all scenarios have equal probability. The general case with unequal probabilities follows by creating more copies of the higher probability scenarios until the distribution is uniform.

A scenario is *covered* when the algorithm decides to stop and choose a value from the opened boxes. For a specific scenario $s \in \mathcal{S}$ we denote by $c(s)$ the total opening cost paid by an algorithm before this scenario is covered and by $v(s)$ the value chosen for this scenario.

**Reservation Values**     To analyze Theorem 3.1, we introduce a new way of defining the reservation values of the boxes that is equivalent to (1). For a box $b$, we have that

$$\sigma_b = \min_{A \subseteq \mathcal{S}} \frac{c_b + \sum_{s \in A} \mathbf{Pr}_{\mathcal{D}}[s] \, v_b^s}{\sum_{s \in A} \mathbf{Pr}_{\mathcal{D}}[s]}$$

The equivalence to (1), follows since $\sigma_b$ is defined as the root of the expression

$$\mathbb{E}_{s \sim \mathcal{D}} \left[ (\sigma_b - v_b^s)^+ \right] - c_b = \sum_{s \in \mathcal{S}} \mathbf{Pr}_{\mathcal{D}}[s] \, (\sigma_b - v_b^s)^+ - c_b$$

$$= \max_{A \subseteq \mathcal{S}} \sum_{s \in A} \mathbf{Pr}_{\mathcal{D}}[s] \, (\sigma_b - v_b^s) - c_b.$$

If we divide the above expression by any positive number, the result will not be affected since we require the root of the equation; $\sigma_b$ being the root is equivalent to $\sigma_b$ being the root of the numerator.

Thus, dividing by $\sum_{s \in A} \mathbf{Pr}_{\mathcal{D}}[s]$ we get that $\sigma_b$ is also the root of

$$\max_{A \subseteq \mathcal{S}} \frac{\sum_{s \in A} \mathbf{Pr}_{\mathcal{D}}[s](\sigma_b - v_b^s) - c_b}{\sum_{s \in A} \mathbf{Pr}_{\mathcal{D}}[s]} = \sigma_b - \min_{A \subseteq \mathcal{S}} \frac{c_b + \sum_{s \in A} \mathbf{Pr}_{\mathcal{D}}[s] v_b^s}{\sum_{s \in A} \mathbf{Pr}_{\mathcal{D}}[s]}. \tag{2}$$

This, gives our formula for computing $\sigma_b$, which we can further simplify using our assumption that all scenarios have equal probability. In this case, $\mathbf{Pr}_{\mathcal{D}}[s] = 1/|\mathcal{S}|$ which implies that

$$\sigma_b = \min_{A \subseteq \mathcal{S}} \frac{c_b|\mathcal{S}| + \sum_{s \in A} v_b^s}{|A|}. \tag{3}$$

## 3.1 Conditioning on $V_b > \sigma_b$

We start by describing the simpler variant of our algorithm where after opening each box we update the distribution by conditioning on the event $V_b > \sigma_b$. This algorithm is *partially adaptive*, since the order for each scenario does not depend on the actual value that is realized every time. At every step the algorithm will either stop or continue opening boxes conditioned on the event "We have not stopped yet" which does not differentiate among the surviving scenarios.

**Theorem 3.2.** *Algorithm 1 is a 4.428-approximation for* PANDORA'S BOX *against the partially-adaptive optimal, when conditioning on* $V_b > \sigma_b$.

In this section we show a simpler proof for Theorem 3.2 that gives a $3 + 2\sqrt{2} \approx 5.828$-approximation. The full proof for the 4.428-approximation is given in section A.2 of the Appendix. Using the equivalent definition of the reservation value (Equation (3)) we can rewrite Algorithm 1 as follows.

---

**Algorithm 2:** Weitzman's rule for Partial Updates

**Input:** Boxes with costs $c_i \in \mathbb{R}$, set of scenarios $\mathcal{S}$.

1   $t \leftarrow 0$
2   $R_0 \leftarrow \mathcal{S}$ the set of scenarios still uncovered
3   **while** $R_t \neq \emptyset$ **do**
4      Let $\sigma_t \leftarrow \min_{b \in \mathcal{B}, A \subseteq R_t} \frac{c_b|R_t| + \sum_{s \in A} v_b^s}{|A|}$
5      Let $b_t$ and $A_t$ be the box and the set of scenarios that achieve the minimum
6      Open box $b_t$ and pay $c_{b_t}$
7      Stop and choose the value $v_{b_t}$ at box $b_t$ if it is less than $\sigma_t$ (see also Fact 3.2.1)
8      Set $c_{b_t} \leftarrow 0$
9      $R_t \leftarrow R_t \setminus A_t$
10     $t \leftarrow t + 1$
11 **end**

---

**Structure of the solution.** An important property to note is that by the equivalent definition of the reservation value (3) the set of scenarios that stop at each step are the ones that give a value at most $\sigma$ for the box opened, as we formally state in the following fact.

**Fact 3.2.1.** *The value at box $b_t$ is less than $\sigma_t$ if and only if $s \in A_t$.*

In equation (8) the set $A_t$ that maximizes the expression contains all the scenarios with value at most $\sigma_b$ for the box $b$. Therefore, the set $A_t$ are exactly the scenarios covered at each step $t$ of the algorithm, and can be removed from consideration.

Before showing our result, observe that this algorithm is partially adaptive; the order of the boxes does not depend on the scenario realized. This holds since we only condition on "not having stopped" (i.e. $\mathcal{D}_{V_b > \sigma_b}$) and therefore each scenario either stops or uses the same updated prior as all other surviving scenarios to calculate the next reservation values. If we were to draw our solution, it would look like a line, (see also Figure 2 in Appendix A.2), which as we observe in Section 3.2 differs from Variant 2.

Moving on to show the proof, we first start by giving a bound on the cost of the algorithm. The cost can be broken down into opening cost plus the value obtained. Since at any time $t$, all remaining

scenarios $R_t$ pay the opening cost $c_{b_t}$, we have that the total opening cost is $\sum_t c_{b_t}|R_t|$. Moreover, the chosen value is given as $\sum_t \sum_{s \in A_t} v_{b_t}^s$. Overall, we have that

$$\text{ALG} = \sum_t \left( c_{b_t}|R_t| + \sum_{s \in A_t} v_{b_t}^s \right) = \sum_t |A_t| \frac{c_{b_t}|R_t| + \sum_{s \in A_t} v_{b_t}^s}{|A_t|} = \sum_t |A_t| \sigma_t.$$

Defining $\sigma_s$ to be the reservation value of scenario $s$ at the time it is covered, i.e. when $s \in A_t$, we get $\text{ALG} = \sum_{s \in \mathcal{S}} \sigma_s$[2]. We follow a *histogram analysis* similar to the proof of Theorem 4 in Feige et al. [2004] for MIN SUM SET COVER and construct the following histograms.

- The $\text{OPT}_o$ histogram: put the scenarios on the x-axis on increasing opening cost order $c_s^{\text{OPT}}$ according to OPT, the height of each scenario is the opening cost it paid.
- The $\text{OPT}_v$ histogram: put the scenarios on the x-axis on increasing covering value order $v_s^{\text{OPT}}$ according to OPT, the height of each scenario is the value with which it was covered.
- The ALG histogram: put scenarios on the x-axis in the order the algorithm covers them. The height of each scenario is $\sigma_s$. Observe that the area of the ALG histogram is exactly the cost of the algorithm.

*Proof of Theorem 3.2.* Initially, observe that the algorithm will eventually stop; every time we open a box we cover at least one scenario (since line 3 is cannot be $\infty$ while scenarios are left uncovered).

To show the approximation factor, we scale the histograms as follows; $\text{OPT}_o$ scale horizontally by $1/\alpha_o$ and vertically by $1/(\beta \cdot \gamma)$, and $\text{OPT}_v$ scale by $1/\alpha_v$ horizontally, for some constants $\alpha_o, \alpha_v, \gamma, \beta \in (0, 1)$ to be determined later[3]. We align the ALG histogram with $\text{OPT}_v$ and $\text{OPT}_o$ so that all of them have the same right-hand side. Observe that the optimal opening cost is the area below the histogram $\text{OPT}_o$ and has increased by $\beta \cdot \gamma \cdot \alpha_o$, and similarly the area below $\text{OPT}_v$ has increased by $\alpha_v$ as a result of the scaling.

To conclude the proof it suffices to show that any point in the ALG histogram is inside the sum of the rescaled $\text{OPT}_v$ and $\text{OPT}_o$ histograms. Consider any point $p$ in the ALG histogram, and let $s$ be its corresponding scenario and $t$ be the time this scenario is covered. We have that the height of the ALG histogram is

$$\sigma_s = \frac{c_{b_t}|R_t| + \sum_{s \in A_t} v_{b_t}^s}{|A_t|} \le \frac{c_b|R_t| + \sum_{s \in A} v_b^s}{|A|} \tag{4}$$

where the last inequality holds for all $A \subseteq R_t$ and any $b \in \mathcal{B}$.

Denote by $c^*$ the opening cost such that $\gamma|R_t|$ of the scenarios in $R_t$ have opening cost less than $c^*$, and by $R_{\text{low}} = \{s \in R_t : c_s^{\text{OPT}} \le c^*\}$ the set of these scenarios. Similarly denote by $v^*$ the value of scenarios in $R_{\text{low}}$ such that $\beta|R_{\text{low}}|$ of the scenarios have value less than $v^*$ and by $L = \{s \in R_{\text{low}} : v_s^{\text{OPT}} \le v^*\}$ these scenarios. This split is shown in Figure 1, and the constants $\beta, \gamma \in (0, 1)$ will be determined at the end of the proof.

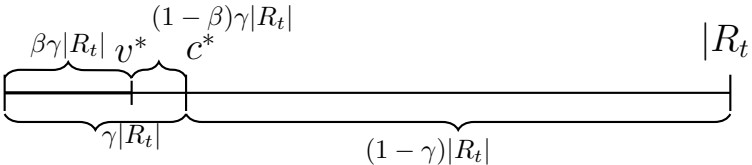

Figure 1: Split of scenarios in $R_t$.

Let $B_L$ be the set of boxes that the optimal solution uses to cover the scenarios in $L$. Let $L_b \subseteq L \subseteq R_t$ be the subset of scenarios in $L$ that choose the value at box $b$ in OPT. Using inequality (4) with

---

[2]Throughout this proof we omit the normalization term $1/|\mathcal{S}|$ both on the algorithms cost and on the optimal cost, without loss of generality, since our guarantee is multiplicative.

[3]Scaling horizontally means that we duplicate every scenario and scaling vertically we just multiply the height at every point by the scale factor.

$b \in B_L$ and $A = L_b$, we obtain $\sigma_s |L_b| \le c_b |R_t| + \sum_{s \in L_b} v_s^{\text{OPT}}$, and by summing up the inequalities for all $b \in B_L$ we get

$$\sigma_s \le \frac{|R_t| \sum_{b \in B_L} c_b + \sum_{s \in L} v_s^{\text{OPT}}}{|L|} \le \frac{|R_t| c^* + \sum_{s \in L} v_s^{\text{OPT}}}{|L|} \le \frac{c^*}{\beta \cdot \gamma} + \frac{\sum_{s \in L} v_s^{\text{OPT}}}{|L|} \tag{5}$$

where for the second inequality we used that the cost for covering the scenarios in $L$ is at most $c^*$ by construction, and in the last inequality that $|L| = |R_t|/(\beta \cdot \gamma)$. We consider each term above separately, to show that the point $p$ is within the histograms.

**Bounding the opening cost.** By the construction of $c^*$, the point in the $\text{OPT}_o$ histogram that has cost at least $c^*$ is at distance at least $(1 - \gamma)|R_t|$ from the right hand side. This means that in the rescaled histogram, the point that has cost at least $c^*/(\beta \cdot \gamma)$ is at distance at least $(1 - \gamma)|R_t|/\alpha_o$ from the right hand side.

On the other hand, in the ALG histogram the distance of $p$ from the right edge of the histogram is at most $|R_t|$, therefore for the point $p$ to be inside the $\text{OPT}_o$ histogram we require

$$\alpha_o \le 1 - \gamma. \tag{6}$$

Observe that throughout the proof we did not use the fact that we change the opening cost to 0, therefore the bound on our cost works even if we re-pay the boxes that are reopened.

The fact that the opening cost becomes 0 is not directly used in the analysis (i.e. inequalities (4) and (5) ). Our analysis gives an upper bound on the cost of the algorithm, even if the algorithm never changes the cost of an opened box to 0. That is the reason in (4) and (5) the cost appears unchanged but the analysis still works for the algorithm since we just want an upper bound (and if we changed the cost to 0 this would only lower the cost of the algorithm).

**Bounding the values cost.** By the construction of $v^*$, the point in the $\text{OPT}_v$ histogram that has value $v^*$ is at distance at least $|R_t|(1 - \beta)\gamma$ from the right hand side. This means that in the rescaled histogram, the point that has value at least $v^*$ is at distance at least $(1 - \beta)\gamma |R_t|/\alpha_v$ from the right hand side.

On the other hand, in the ALG histogram the distance of $p$ from the right edge of the histogram is at most $|R_t|$, therefore for the point $p$ to be inside the $\text{OPT}_o$ histogram we require

$$\alpha_v \le (1 - \beta)\gamma. \tag{7}$$

We optimize the constants $\alpha_o, \alpha_v, \beta, \gamma$ by ensuring that inequalities (6) and (7) hold. We set $\alpha_o = 1 - \gamma$ and $\alpha_v = (1 - \beta)\gamma$, and obtain that $\text{ALG} \le \text{OPT}_o/(\beta \cdot \gamma \cdot (1 - \gamma)) + \text{OPT}_v/((1 - \beta)\gamma)$. Requiring these to be equal we get $\beta = 1/(2 - \gamma)$, which is minimized for $\beta = 1/\sqrt{2}$ and $\gamma = 2 - \sqrt{2}$ for a value of $3 + 2\sqrt{2}$.

$\square$

### 3.2 Conditioning on $V_b = v$

In this section we switch gears to our second variant of Algorithm 1, where in each step we update the prior $\mathcal{D}$ conditioning on the event $V_b = v$. We state our result in Theorem 3.3. In this case, the conditioning on $\mathcal{D}$ implies that the algorithm at every step removes the scenarios that are *inconsistent* with the value realized. In order to understand better the differences of the two variants and their conditioning we included an example and a discussion in section A.1 of the Appendix.

**Theorem 3.3.** *Algorithm 1 is a $3 + 2\sqrt{2} \approx 5.828$-approximation for* PANDORA'S BOX *against the partially-adaptive optimal, when conditioning on $V_b = v$.*

The main challenge was that the algorithm's solution is now a tree with respect to scenarios instead of a line as in the case of $\mathcal{D}|_{V_b > \sigma_b}$. Specifically, in the $D|_{V_b > \sigma_b}$ variant at every step all scenarios that had $V_b \le \sigma_b$ were covered and removed from consideration. However in the $D|_{V_b = v}$ variant the remaining scenarios are split into different cases, based on the realization of $V$, as shown in the example of Figure 4, which is deferred to Section A.3 of the Appendix due to space constraints.

This results into the ALG histogram not being well defined, since there is no unique order of covering the scenarios. We overcome this by generalizing the histogram approach to trees.

*Proof of Theorem 3.3.* The proof follows similar steps to that of Theorem 3.2, thus we only highlight the differences. The algorithm is presented below, the only change is line 5 where we remove the inconsistent with the value revealed scenarios, which also leads to our solution branching out for different scenarios and forming a tree.

---

**Algorithm 3:** Weitzman's rule for Full Updates

---

**Input:** Boxes with costs $c_i \in \mathbb{R}$, set of scenarios $\mathcal{S}$.

1 Define a root node $u$ corresponding to the set $\mathcal{S}$
2 $R_u \leftarrow \mathcal{S}$ the set of scenarios still uncovered
3 **while** $R_u \neq \emptyset$ **do**
4 $\quad$ Let $\sigma_u \leftarrow \min_{b \in \mathcal{B}, A \subseteq R_u} \frac{c_b |R_u| + \sum_{s \in A} v_b^s}{|A|}$
5 $\quad$ Let $b_u$ and $A_u$ be the box and the set of scenarios that achieve the minimum
6 $\quad$ Open box $b_u$ paying $c_{b_u}$ and observe value $v$
7 $\quad$ Stop and choose the value at box $b_u$ if it is less than $\sigma_u$: this holds **iff** $s \in A_u$
8 $\quad$ Set $c_{b_u} \leftarrow 0$
9 $\quad$ Let $u'$ be a vertex corresponding to the set of consistent scenarios with
$\qquad R_{u'} \triangleq R_u \setminus \left( A_u \cup \{s \in R_u : v_{b_u}^s \neq v\} \right)$ // Remove inconsistent scenarios
10 $\quad$ Set $u \leftarrow u'$
11 **end**

---

**Bounding the opening cost** Consider the tree $\mathcal{T}$ of ALG where at every node $u$ a set $A_u$ of scenarios is covered. We associate this tree with node weights, where at every node $u$, we assign $|A_u|$ weights $(\sigma_u, ..., \sigma_u)$. Denote, the weighted tree by $\mathcal{T}_{\text{ALG}}$. As before, the total cost of ALG is equal to the sum of the weights of the tree.

We now consider two alternative ways of assigning weights to the the nodes, forming trees $\mathcal{T}_{\text{OPT}_o}$, $\mathcal{T}_{\text{OPT}_v}$ using the following process.

- $\mathcal{T}_{\text{OPT}_o}$. At every node $u$ we create a vector of weights $\boldsymbol{w}_u^{\text{OPT}_o} = (c_s^{\text{OPT}})_{s \in A_u}$ where each $c_s^{\text{OPT}}$ is the opening cost that scenario $s \in A_u$ has in the optimal solution.

- $\mathcal{T}_{\text{OPT}_v}$. At every node $u$ we create a vector of weights $\boldsymbol{w}_u^{\text{OPT}_v} = (v_s^{\text{OPT}})_{s \in A_u}$ where each $v_s^{\text{OPT}}$ is the value the optimal uses to cover scenario $s \in A_u$.

We denote by $\text{cost}(\mathcal{T}_{\text{ALG}})$ the sum of all weights in every node of the tree $\mathcal{T}$. We have that $\text{cost}(\mathcal{T})$ is equal to the total cost of ALG, while $\text{cost}(\mathcal{T}_{\text{OPT}_o})$ and $\text{cost}(\mathcal{T}_{\text{OPT}_v})$ is equal to the optimal opening cost $\text{OPT}_o$ and optimal value $\text{OPT}_v$ respectively. Intuitively, the weighted trees correspond to the histograms in the previous analysis of Theorem 3.2.

We want to relate the cost of ALG, to that of $\mathcal{T}_{\text{OPT}_o}$ and $\mathcal{T}_{\text{OPT}_v}$. To do this, we define an operation similar to histogram scaling, which replaces the weights of every node $u$ in a tree with the top $\rho$-percentile of the weights in the subtree rooted at $u$. As the following lemma shows, this changes the cost of a tree by a bounded multiplicative factor.

**Lemma 3.3.1.** *Let $\mathcal{T}$ be a tree with a vector of weights $\boldsymbol{w}_u$ at each node $u \in \mathcal{T}$, and let $\mathcal{T}^{(\rho)}$ be the tree we get when we substitute the weights of every node with the top $\rho$-percentile of all the weights in the subtree of $\mathcal{T}$ rooted at $u$. Then*

$$\rho \cdot \text{cost}(\mathcal{T}^{(\rho)}) \leq \text{cost}(\mathcal{T}).$$

We defer the proof of Lemma 3.3.1 to Section A.3 of the Appendix. To complete the proof of Theorem 3.3, and bound $\text{cost}(\mathcal{T}_{\text{ALG}})$, we show as before that the weights at every node $u$, are bounded by the weights of $\mathcal{T}_{\text{OPT}_o}^{(1-\gamma)}$ scaled by $\frac{1}{\beta\gamma}$ plus the weights of $\mathcal{T}_{\text{OPT}_v}^{((1-\beta)\gamma)}$, for the constants $\beta, \gamma \in (0, 1)$ chosen in the proof of Theorem 3.2. This implies that

$$\text{cost}(\mathcal{T}_{\text{OPT}_o}) \leq \frac{1}{\beta\gamma} \text{cost}(\mathcal{T}_{\text{OPT}_o}^{(1-\gamma)}) + \text{cost}(\mathcal{T}_{\text{OPT}_v}^{((1-\beta)\gamma)})$$

$$\leq \frac{1}{\beta\gamma(1\text{-}\gamma)}\mathrm{cost}(\mathcal{T}_{\mathrm{OPT}_o}) + \frac{1}{(1\text{-}\beta)\gamma}\mathrm{cost}(\mathcal{T}_{\mathrm{OPT}_v})$$

which gives ALG $\leq 5.828$ OPT for the choice of $\beta$ and $\gamma$. The details of the proof are similar to the one of Theorem 3.1, and are deferred to section A.3 of the Appendix.

□

**Note on the approximation factors.** Observe that Variant 2, where we condition on $V_b = v$ has a worse approximation factor than Variant 1 where we only condition on $V_b > \sigma_b$. Intuitively someone might expect that with more information the approximation factor will improve. However, it is challenging to argue about this formally. It is also plausible that such monotonicity may not hold as more information might lead the greedy algorithm to make wrong decisions. Instead of making any such claims, we analyze this case directly by showing that our proof approach extends to the full update variant with a generalization of the histogram method to work on trees. Our technique for improving the approximation for the partial updates variant could not be generalized however and thus we only obtain the worse approximation guarantee.

## 4 Learning from Samples

In this section we show that our algorithm also works when we are only given sample access to the correlated distribution $\mathcal{D}$.

We will mainly focus on the first variant with partial updates $\mathcal{D}|_{V>v}$. The second variant with full Bayesian updates $\mathcal{D}|_{V=v}$ requires full knowledge of the underlying distribution and can only work with sample access if one can learn the full distribution. To see this consider for example an instance where the values are drawn uniformly from $[0,1]^d$. No matter how many samples one draws, it is impossible to know the conditional distribution $\mathcal{D}|_{V=v}$ after opening the first box for fresh samples $v$, and the Bayesian update is not well defined[4].

Variant 1 does not face this problem and can be learned from samples if the costs of the boxes are polynomially bounded by $n$, i.e. if there is a constant $c > 0$ such that for all $b \in \mathcal{B}$, $c_b \in [1, n^c]$. If the weights are unbounded, it is impossible to get a good approximation with few samples. To see this consider the following instance. Box 1 has cost $1/H \to 0$, while every other box has cost $H$ for a very large $H > 0$. Now consider a distribution where with probability $1 - \frac{1}{H} \to 1$, the value in the first box is $0$, and with probability $1/H$ is $+\infty$. In this case, with a small number of samples we never observe any scenario where $v_1 \neq 0$ and believe the overall cost is near $0$. However, the true cost is at least $H \cdot 1/H \geq$ and is determined by how the order of boxes is chosen when the scenario has $v_1 \neq 0$. Without any such samples it is impossible to pick a good order.

Therefore, we proceed to analyze Variant 1 with $\mathcal{D}|_{V>\sigma}$ in the case when the box costs are similar. We show that polynomial, in the number of boxes, samples suffice to obtain an approximately-optimal algorithm, as we formally state in the following theorem. We present the case where all boxes have cost 1 but the case where the costs are polynomially bounded easily follows.

**Theorem 4.1.** *Consider an instance of Pandora's Box with opening costs equal to 1. For any given parameters $\varepsilon, \delta > 0$, using $m = poly(n, 1/\varepsilon, \log(1/\delta))$ samples from $\mathcal{D}$, Algorithm 1 (Variant 1) obtains a $4.428 + \varepsilon$ approximation policy against the partially-adaptive optimal, with probability at least $1 - \delta$.*

To prove the theorem, we first note that variant 1 of Algorithm 1 takes a surprisingly simple form, which we call a threshold policy. It can be described by a permutation $\pi$ of visiting the boxes and a vector of thresholds $\boldsymbol{\tau}$ that indicate when to stop. The threshold for every box corresponds to the reservation value the first time the box is opened. To analyze the sample complexity of Algorithm 1, we study a broader class of algorithms parameterized by a permutation and vector of thresholds given in Algorithm 4.

Our goal now is to show that polynomially many samples from the distribution $\mathcal{D}$ suffice to learn good parameters for Algorithm 4. We first show a Lemma that bounds the cost of the algorithm calculated in the empirical $\hat{\mathcal{D}}$ instead of the original $\mathcal{D}$ (Lemma 4.1.1), and a Lemma 4.1.2 that shows how capping the reservation values by $n/\varepsilon$ can also be done with negligible cost.

---

[4]For a discrete distribution example see Section A.5 of the appendix.

---

**Algorithm 4:** General format of PANDORA'S BOX algorithm.

---
**Input:** Set of boxes, permutation $\pi$, vector of thresholds $\boldsymbol{\tau} \in \mathbb{R}^n$

1   best $\leftarrow \infty$
2   **foreach** $i \in [n]$ **do**
3      **if** *best* $> \tau_i$ **then**
4         Open box $\pi_i$, see value $v_i$
5         best $\leftarrow \min(\text{best}, v_i)$
6      **else**
7         Accept best
8   **end**

---

**Lemma 4.1.1.** *Let $\varepsilon, \delta > 0$ and let $\mathcal{D}'$ be the empirical distribution obtained from $poly(n, 1/\varepsilon, \log(1/\delta))$ samples from $\mathcal{D}$. Then, with probability $1 - \delta$, it holds that*

$$\left| \mathbb{E}_{\hat{D}} \left[ ALG(\pi, \tau) - \min_{b \in \mathcal{B}} v_b \right] - \mathbb{E}_D \left[ ALG(\pi, \tau) - \min_{b \in \mathcal{B}} v_b \right] \right| \leq \varepsilon$$

*for any permutation $\pi$ and any vector of thresholds $\boldsymbol{v} \in \left[ 0, \frac{n}{\varepsilon} \right]^n$*

We defer the proof of Lemmas 4.1.1, 4.1.2 and that of Theorem 4.1 to Section A.5 of the Appendix.

**Lemma 4.1.2.** *Let $\mathcal{D}$ be any distribution of values. Let $\varepsilon > 0$ and consider a permutation $\pi$ and thresholds $\boldsymbol{\tau}$. Moreover, let $\tau'$ be the thresholds capped to $n/\varepsilon$, i.e. setting $\tau'_b = \min\{\tau_b, n/\varepsilon\}$ for all boxes $b$. Then,*

$$\mathbb{E}_{v \sim D} [ALG(\pi, \tau')] \leq (1 + \varepsilon) \mathbb{E}_{v \sim D} [ALG(\pi, \tau)].$$

**Note on Continuous vs Discrete Distributions.** The results of Section 4 apply for general distributions (discrete or continuous) and show that the partial updates variant leads to good approximation when run on the empirical distribution obtained just with polynomially many samples. In contrast, the full updates variant requires a complete description of the distribution. However, as the approximation factor does not depend on the support size, It can also apply even for continuous distributions with arbitrary large support by taking a limit over a very fine discretization

## 5   Conclusion

We present a summary of our results with a comparison to previous work on Table 1. Our main contribution was to improve the approximation factor for Pandora's Box with correlations given by Chawla et al. [2020], while also greatly simplifying their approach. Our algorithm also directly extends the independent case algorithm, giving us a unified way to solve this problem. An interesting open question is to try and improve their results for more complex combinatorial constraints, like selecting $k$ boxes (instead of one) or for selecting a basis of size $k$, when the boxes are part of a matroid.

| | Approx. Factor | Learnable from Samples |
|---|---|---|
| Algorithm of Chawla et al. [2020] | 9.22 | Yes |
| **Variant 1** ($D_{V_b > \sigma_b}$) | **4.428** (Thm 3.2) | **Yes** (Thm 4.1 ) |
| **Variant 2** ($D_{V_b = v}$) | **5.828** (Thm 3.3) | **No** (Sec. 4) |

Table 1: Summary of our results (in bold) and comparison to previous work.

Observe also that the more natural Variant 2 seems worse than Variant 1 even though the algorithm has more accurate information through the update of the prior. Intuitively we would expect a better factor, however since the algorithm is greedy approximation, and not the optimal, the factor may not necessarily be monotone on the amount of information given. We leave as an open problem whether our analysis in Variant 2 is tight or this greedy algorithm cannot perform better under full information.

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

# A Appendix

## A.1 Supplemental Preliminaries

**Updating the prior.** We include an example showing the process of updating the prior for our two different updating rules. The (correlated) distribution is a set of vectors of size $n$, where each is drawn with some probability. When we open a box and see a value, some scenarios are not "possible" anymore, i.e. we know they cannot be the ones realized. We illustrate in the following example. Assume there are 3 of these vectors (scenarios).

|       | $b_1$ | $b_2$ | $b_3$ |
|-------|-------|-------|-------|
| $S_1$ | 3     | 4     | 7     |
| $S_2$ | 6     | 4     | 2     |
| $S_3$ | 7     | 7     | 2     |

Table 2: Example with 3 scenarios and 3 boxes.

The rows in the matrix above are the scenarios, and the columns are the boxes. For example, if scenario $S_2$ is the one realized (i.e. drawn from the distribution) then the values inside boxes $b_1, b_2$ and $b_3$ are 6, 4 and 2 respectively. The distribution $\mathcal{D}$ is essentially drawing one of the scenarios with some probability.

To see what the conditioning means: assume we open box $b_1$ and we see the value 6 (and assume for the sake of the example that the reservation value of box 1 is $\sigma_1 = 5$).

- **Variant 1**: we condition on $6 = V_b > \sigma_1 = 5$ meaning that scenario $S_1$ is not possible anymore (because if $S_1$ was the one drawn from $\mathcal{D}$, then we would have seen a value less than $\sigma_1 = 5$ when opening the box), and is removed from the set S the algorithm considers (line 9, Alg 2)
- **Variant 2**: we condition on $V_b = 6$, which means that scenarios $S_1$ and $S_3$ are both removed (similarly, because if any of these were drawn, we would not have seen 6 upon opening the box)

**Differences in the variants.** As a result of the different conditioning, the solution for the $V_b > \sigma$ variant is *partially adaptive* meaning that the next box the algorithm opens, only depends on the scenarios that remain. However, for the $V_b = v$ variant the solution is *fully adaptive* (meaning that the next box opened, depends on the exact value seen). This is illustrated in Figures 2 and 4 in the Appendix, where Variant 1's solution can be represented by a line graph (Figure 2), while Variant 2's solution is a tree (Figure 4).

## A.2 Proofs from Section 3

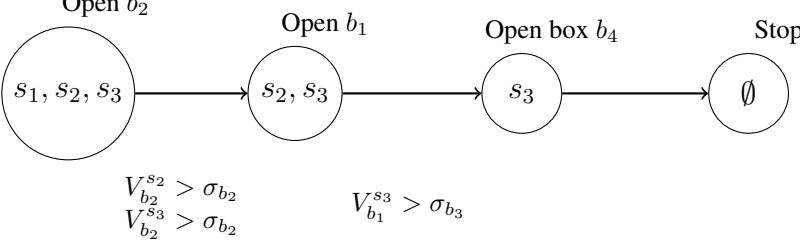

Figure 2: Algorithm's solution when $\mathcal{D} \leftarrow \mathcal{D}_{V>\sigma}$, for an instance with scenarios $\mathcal{S} = \{s_1, s_2, s_3\}$, and boxes $\mathcal{B} = \{b_1, b_2, b_3, b_4\}$. The circles contain the scenarios that have not stopped at each step. Scenario $s_1$ stopped at box $b_2$, scenario $s_2$ stopped at box $b_1$ and $s_3$ at box $b_4$.

**Theorem 3.2.** *Algorithm 1 is a* 4.428-*approximation for* PANDORA'S BOX *against the partially-adaptive optimal, when conditioning on* $V_b > \sigma_b$.

The tighter guarantee proof follows the steps of the proof in section 3.1 for the opening cost, but provides a tighter analysis for the values cost.

*Tight proof of Theorem 3.2.* Denote by $\sigma_s$ the reservation value for scenario $s$ when it was covered by ALG and by $\mathcal{T}$ the set of boxes opened i.e. the steps taken by the algorithm. Then we can write the cost paid by the algorithm as follows

$$\text{ALG} = \frac{1}{|\mathcal{S}|} \sum_{s \in \mathcal{S}} \sigma_s = \frac{1}{|\mathcal{S}|} \sum_{p \in \mathcal{T}} |A_t| \sigma_p. \tag{8}$$

We use the same notation as section 3.1 which we repeat here for convenience. Consider any point $p$ in the ALG histogram, and let $s$ be its corresponding scenario and $t$ be the time this scenario is covered.

- $R_t$ : set of uncovered scenarios at step $t$
- $A_t$ : set of scenarios that ALG chooses to cover at step $t$
- $c^*$: the opening cost such that $\gamma |R_t|$ of the scenarios in $R_t$ have opening cost less than $c^*$
- $R_{\text{low}} = \{s \in R_t : c_s^{\text{OPT}} \leq c^*\}$ the set of these scenarios
- $v^*$: the value of scenarios in $R_{\text{low}}$ such that $b|R_{\text{low}}|$ of the scenarios have value less than $v^*$
- $L = \{s \in R_{\text{low}} : v_s^{\text{OPT}} \leq v^*\}$ the set of scenarios with value at most $v^*$
- $B_L$: set of boxes the optimal uses to cover the scenarios in $L$ of step $t$

The split described in the definitions above is again shown in Figure 3, and the constants $1 > \beta, \gamma > 0$ will be determined in the end of the proof.

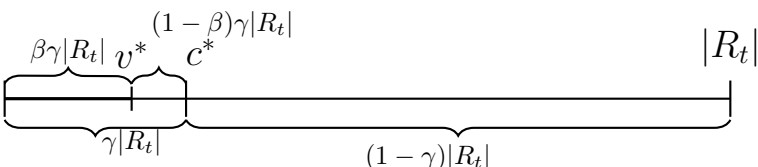

Figure 3: Split of scenarios in $R_t$.

Continuing from equation (8) we obtain the following.

$$\text{ALG} \leq \frac{1}{|\mathcal{S}|} \sum_{t \in \mathcal{T}} |A_t| \frac{|R_t| \sum_{b \in B_L} c_b + \sum_{s \in L} v_s^{\text{OPT}}}{|L|} \qquad \text{Inequality 5}$$

$$\leq \frac{1}{|\mathcal{S}|} \sum_{t \in \mathcal{T}} \left( |A_t| \frac{c^*}{\beta\gamma} + \frac{\sum_{s \in L} v_s^{\text{OPT}}}{|L|} \right) \qquad \text{Ineq. 5 and } |L| = \gamma\beta|R_t|$$

$$\leq \frac{\text{OPT}_o}{\beta\gamma(1-\gamma)} \sum_{t \in \mathcal{T}} \frac{|A_t|}{|\mathcal{S}|} + \sum_{t \in \mathcal{T}} \frac{|A_t|}{|\mathcal{S}|} \frac{\sum_{s \in L} v_s^{\text{OPT}}}{|L|} \qquad \text{Since } c^* \leq \text{OPT}_o/(1-\gamma)$$

$$= \frac{\text{OPT}_o}{\beta\gamma(1-\gamma)} + \sum_{p \in \mathcal{T}} \frac{|A_t|}{|\mathcal{S}|} \frac{\sum_{s \in L} v_s^{\text{OPT}}}{|L|} \qquad \text{Since } \sum_t |A_t| = |\mathcal{S}|$$

Where in the second to last inequality we used the same histogram argument from section 3.1, to bound $c^*$ by $\text{OPT}_o/(1-\gamma)$.

To bound the values term, observe that if we sorted the optimal values $v_s^{\text{OPT}}$ that cover each scenario by decreasing order, and denote $j_s$ the index of $v_s^{\text{OPT}}$ in this ordering, we add $v_s^{\text{OPT}}$ multiplied by

the length of the interval every time $j_s \in \big[(1-\beta)\gamma|R_t|, \gamma|R_t|\big]$. This implies that the length of the intervals we sum up for $v_s^{\text{OPT}}$ ranges from $j_s/\gamma$ to $j_s/((1-\beta)\gamma)$, therefore the factor for each $v_s^{\text{OPT}}$ is

$$\frac{1}{\gamma} \sum_{i=j_s/\gamma}^{j_s/(1-\beta)\gamma} \frac{1}{i} \leq \frac{1}{\gamma} \log\left(\frac{1}{1-\beta}\right)$$

We want to balance the terms $1/(\beta\gamma(1-\gamma))$ and $1/\gamma \log(1/(1-\beta))$ which gives that

$$\gamma = 1 - \frac{1}{\beta \log\left(\frac{1}{1-\beta}\right)}.$$

Since we balanced the opening cost and value terms, by substituting the expression for $\gamma$ we get that the approximation factor is

$$\frac{1}{\beta\gamma(1-\gamma)} = \frac{\beta \log^2\left(\frac{1}{1-\beta}\right)}{\beta \log\left(\frac{1}{1-\beta}\right) - 1}.$$

Numerically minimizing that ratio for $\beta$ and ensuring that $0 < \beta, \gamma < 1$ we get that the minimum is 4.428 obtained at $\beta \approx 0.91$ and $\gamma \approx 0.55$. $\qquad\square$

### A.3  Proofs from Section 3.2

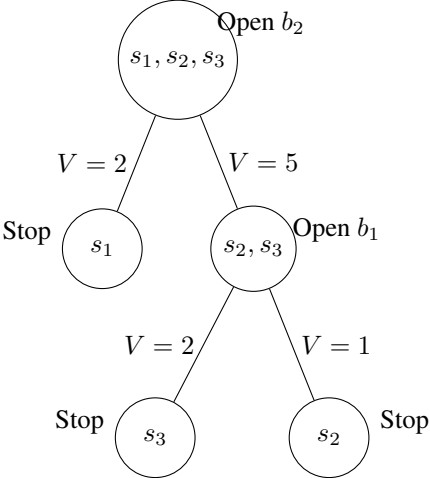

Figure 4: Algorithm's solution when conditioning on $V = v$, for an instance with scenarios $\mathcal{S} = \{s_1, s_2, s_3\}$, and boxes $\mathcal{B} = \{b_1, b_2\}$. The nodes contain the consistent scenarios at each step, and the values $V$ are revealed once we open the corresponding box.

**Theorem 3.3.** *Algorithm 1 is a $3 + 2\sqrt{2} \approx 5.828$-approximation for* PANDORA'S BOX *against the partially-adaptive optimal, when conditioning on $V_b = v$.*

*Continued proof of Theorem 3.3.* We now proceed to give the bound on the weights of the nodes of $\mathcal{T}_{\text{ALG}}$. Consider any node $u$. We have that the weights at this node are equal to

$$\sigma_u = \frac{c_{b_u}|R_u| + \sum_{s \in A_t} v_{b_u}^s}{|A_t|} \leq \frac{c_b|R_u| + \sum_{s \in A} v_b^s}{|A|}$$

where the last inequality holds for all $A \subseteq R_u$ and any $b \in \mathcal{B}$.

Let $c_u^*$ be the opening cost such that $\gamma|R_u|$ of the scenarios in $R_u$ have opening cost less than $c_u^*$, and by $R_{\text{low}} = \{s \in R_u : c_s^{\text{OPT}} \leq c_u^*\}$ the set of these scenarios. Similarly denote by $v_u^*$ the value of scenarios in $R_{\text{low}}$ such that $\beta|R_{\text{low}}|$ of the scenarios have value less than $v_u^*$ and by $L = \{s \in R_{\text{low}}^p : v_s^{\text{OPT}} \leq v_u^*\}$ these scenarios. This split is shown in Figure 1.

Note that, $c_u^*$ corresponds to the weights of node $u$ in $\mathcal{T}_{\text{OPT}_o}^{(1-\gamma)}$, while the weights of node $u$ at $\mathcal{T}_{\text{OPT}_v}^{(1-\gamma)}$ are at least $v_u^*$.

Let $B_L$ be the set of boxes that the optimal solution uses to cover the scenarios in $L$. Let $L_b \subseteq L \subseteq R_u$ be the subset of scenarios in $L$ that choose the value at box $b$ in OPT. Using inequality (4) with $b \in B_L$ and $A = L_b$, we obtain $\sigma_u|L_b| \le c_b|R_u| + \sum_{s \in L_b} v_s^{\text{OPT}}$, and by summing up the inequalities for all $b \in B_L$ we get

$$\sigma_u \le \frac{|R_u|\sum_{b \in B_L} c_b + \sum_{s \in L} v_s^{\text{OPT}}}{|L|} \le \frac{|R_u|c^* + \sum_{s \in L} v_s^{\text{OPT}}}{|L|} \le \frac{c_u^*}{\beta \cdot \gamma} + v_u^* \qquad (9)$$

where for the second inequality we used that the cost for covering the scenarios in $L$ is at most $c_u^*$ by construction, and in the last inequality that $|L| = |R_t|/(\beta \cdot \gamma)$. We consider each term above separately, to show that the point $p$ is within the histograms. $\qquad \square$

**Lemma 3.3.1.** *Let $\mathcal{T}$ be a tree with a vector of weights $\boldsymbol{w}_u$ at each node $u \in \mathcal{T}$, and let $\mathcal{T}^{(\rho)}$ be the tree we get when we substitute the weights of every node with the top $\rho$-percentile of all the weights in the subtree of $\mathcal{T}$ rooted at $u$. Then*

$$\rho \cdot cost(\mathcal{T}^{(\rho)}) \le cost(\mathcal{T}).$$

*Proof of Lemma 3.3.1.* We denote by $\mathcal{T}_u$ the subtree rooted at $u$, by $W(\mathcal{T}) = \{w : w \in \boldsymbol{w}_v$ for $v \in \mathcal{T}\}$ the (multi)set of weights in the tree $\mathcal{T}$. Denote, by $q^\rho(\mathcal{T})$ be the top $\rho$ percentile of all the weights in $\mathcal{T}$. Finally, we define $Q(\rho|\mathcal{T})$ for any tree $\mathcal{T}$ as follows:

- We create a histogram $H(x)$ of the weights in $W(\mathcal{T})$ in increasing order.

- We calculate the area enclosed within $(1-\rho)|W(\mathcal{T})|$ until $|W(\mathcal{T})|$:

$$Q(\rho|\mathcal{T}) = \int_{(1-\rho)|W(\mathcal{T})|}^{|W(\mathcal{T})|} H(x)dx$$

  This is approximately equal to the sum of all the values greater than $q^\rho(\mathcal{T})$ with values exactly $q^\rho(\mathcal{T})$ taken fractionally so that exactly $\rho$ fraction of values are selected.

We show by induction that for every node $u$, it holds that $\rho \cdot \text{cost}(\mathcal{T}_u^{(\rho)}) \le Q(\rho|\mathcal{T})$

- For the base case, for all leaves $u$, the subtree $\mathcal{T}_u$ only has one node and the lemma holds as $\rho q^\rho(\mathcal{T}_u) \le Q(\rho|\mathcal{T}_u)$.

- Now, let $r$ be any node of the tree, and denote by $\text{child}(r)$ the set of the children nodes of $r$.

$$\rho \cdot \text{cost}(\mathcal{T}_r^{(\rho)}) = \rho \cdot q^\rho(\mathcal{T}_r)|\boldsymbol{w}_r| + \rho \cdot \sum_{v \in \text{child}(r)} \text{cost}(\mathcal{T}_v^{(\rho)}) \qquad \text{Definition of cost}(\mathcal{T}_r^{(\rho)})$$

$$\le \rho \cdot q^\rho(\mathcal{T}_r)|\boldsymbol{w}_r| + \rho \cdot \sum_{v \in \text{child}(r)} Q(\rho|T_v) \qquad \text{From induction hypothesis}$$

$$\le \rho \cdot q^\rho(\mathcal{T}_r)|\boldsymbol{w}_r| + Q\left(\rho \frac{|W(\mathcal{T}_r)| - |\boldsymbol{w}_r|}{|W(\mathcal{T}_r)|} \middle| T_r\right) \qquad \text{Since } \mathcal{T}_v \subseteq T_r$$

$$\le Q(\rho|T_r)$$

The second-to-last inequality follows since $Q$ is defined as the area of the largest weights of the histogram. Including more weights only increases and keeping the length of the integration range the same (equal to $\rho(|W(\mathcal{T}_r)| - |\boldsymbol{w}_r|)$) can only increase the value $Q$.

The last inequality follows by noting that if $H(x)$ is the histogram corresponding to the values of $\mathcal{T}_r$, then

$$Q(\rho|T_r) - Q\left(\rho \frac{|W(\mathcal{T}_r)| - |\boldsymbol{w}_r|}{|W(\mathcal{T}_r)|} \middle| T_r\right) = \int_{(1-\rho)|W(\mathcal{T}_r)|}^{|W(\mathcal{T}_r)|} H(x)dx - \int_{(1-\rho)|W(\mathcal{T}_r)|+\rho|\boldsymbol{w}_r|}^{|W(\mathcal{T}_r)|} H(x)dx$$

$$= \int_{(1-\rho)|W(\mathcal{T}_r)|}^{(1-\rho)|W(\mathcal{T}_r)|+\rho|\boldsymbol{w}_r|} H(x)dx \geq \int_{(1-\rho)|W(\mathcal{T}_r)|}^{(1-\rho)|W(\mathcal{T}_r)|+\rho|\boldsymbol{w}_r|} q^\rho(\mathcal{T}_r)dx$$
$$= \rho q^\rho(\mathcal{T}_r)|\boldsymbol{w}_r|$$

where the inequality follows since $H(x) \geq q^\rho(\mathcal{T}_r)$ for $x \geq (1-\rho)|W(\mathcal{T}_r)|$ by the definition of $q^\rho(\mathcal{T}_r)$ as the top-$r$ quantile of the weights in $\mathcal{T}_r$.

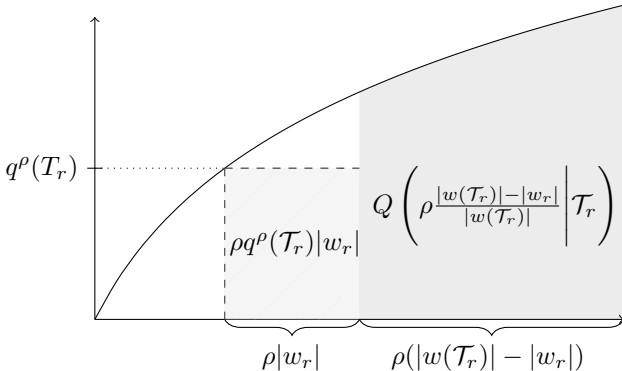

Figure 5: Picture depicting the proof above.

$\square$

## A.4 Lower Bound

To show that our algorithm is almost tight, we observe that the lower bound of Min Sum Set Cover presented in Feige et al. [2004] also applies to PANDORA'S BOX. In MIN SUM SET COVER we are given $n$ elements $e_i$, and $m$ sets $s_j$ where each $s_j \subseteq [n]$. We say a set $s_j$ *covers* an element $e_i$ if $e_i \in s_j$. The goal is to select elements in order to minimize the sum of the *covering times* of all the sets, where *covering time* of a set is the first time an element $e_i \in s_j$ is chosen. This lower bound is also mentioned in Chawla et al. [2020], but we include it here with more details for the sake of completeness.

In Feige et al. [2004] the authors show that MIN SUM SET COVER cannot be approximated better than $4 - \varepsilon$ even in the special case where every set contains the same number of elements[5]. We restate the theorem below.

**Theorem A.1** (Theorem 13 of Feige et al. [2004]). *For every $\varepsilon > 0$, it is NP-hard to approximate min sum set cover within a ratio of $4 - \varepsilon$ on uniform hypergraphs.*

Our main observation is that MIN SUM SET COVER is a special case of PANDORA'S BOX. When the boxes all have the same opening cost $c_b = 1$ and the values inside are $v_s^b \in \{0, \infty\}$, we are required to find a $0$ for each scenario; equivalent to *covering* a scenario. The optimal solution of MIN SUM SET COVER is an algorithm that selects elements one by one, and stops whenever all the sets are covered. This is exactly the partially adaptive optimal we defined for PANDORA'S BOX. The theorem restated above results in the following Corollary.

**Corollary A.1.1.** *For every $\varepsilon > 0$ it is NP-Hard to approximate Pandora's Box against the partially-adaptive within a ratio better than $4 - \varepsilon$.*

## A.5 Proofs from Section 4

We first present an example of a discrete distribution that shows that one needs exponentially many samples in the number of boxes to learn $D_{V=v}$.

**Discrete Distribution Example** Consider a distribution that only takes values $0, H, H + 1$ for some very large $H > 0$. The scenario is drawn by choosing a random bit $b_i \in \{0, 1\}$ for every box

---

[5]Equivalently forms a uniform hypergraph, where sets are hyperedges, and elements are vertices.

and depending on the realized sequence $\boldsymbol{b}$ a single box $f(\boldsymbol{b}) \in [n]$ is chosen for an unknown and arbitrary function $f$. The value at box $i$ is then chosen to be $H + b_i$ unless $i$ is the box $f(\boldsymbol{b})$ in which case it is 0. In this case learning the probability $D_{V=v}$ would require learning the unknown function $f$ on all inputs which are exponentially many. In particular, if we only take $s << 2^n$ samples, for any order of choosing boxes after $\approx \log s$ steps, none of the samples in our collection will match the observed sequence of bits, therefore it will not be possible to compute a posterior distribution.

We continue by giving the omitted proofs.

**Lemma 4.1.1.** *Let $\varepsilon, \delta > 0$ and let $\mathcal{D}'$ be the empirical distribution obtained from $\mathrm{poly}(n, 1/\varepsilon, \log(1/\delta))$ samples from $\mathcal{D}$. Then, with probability $1 - \delta$, it holds that*

$$\left| \mathbb{E}_{\hat{D}}\left[ ALG(\pi, \tau) - \min_{b \in \mathcal{B}} v_b \right] - \mathbb{E}_D\left[ ALG(\pi, \tau) - \min_{b \in \mathcal{B}} v_b \right] \right| \leq \varepsilon$$

*for any permutation $\pi$ and any vector of thresholds $\boldsymbol{v} \in \left[0, \frac{n}{\varepsilon}\right]^n$*

*Proof of Lemma 4.1.1.* We first argue that we can accurately estimate the cost for any vector of thresholds $\boldsymbol{\tau}$ when the order of visiting boxes is fixed.

Consider any fixed permutation $\pi = \pi_1, \pi_2, \ldots, \pi_n$ be any permutation of the boxes, we relabel the boxes without loss of generality so that $\pi_i$ is box $i$.

Denote by $\hat{V}_i = \min_{j \leq i} v_j$, and observe that $\hat{V}_i$ is a random variable that depends on the distribution $\mathcal{D}$. Then we can write the expected cost of the algorithm as the expected sum of the opening cost and the chosen value: $\mathbb{E}_{\mathcal{D}}[\mathrm{ALG}] = \mathbb{E}_{\mathcal{D}}[\mathrm{ALG}_o] + \mathbb{E}_{\mathcal{D}}[\mathrm{ALG}_v]$. We have that:

$$\mathbb{E}_{\mathcal{D}}[\mathrm{ALG}_o] = \sum_{i=1}^{n} \mathbf{Pr}_{\mathcal{D}}[\text{reach } i] = \sum_{i=1}^{n} \mathbf{Pr}_{\mathcal{D}}\left[ \bigwedge_{j=1}^{i-1}(\hat{V}_j > \tau_{j+1}) \right]$$

Moreover, we denote by $\overline{V}^i_{\boldsymbol{\tau}} = \bigwedge_{j=1}^{i-1}\left(\hat{V}_j > \tau_{j+1}\right)$ and we have

$$\mathbb{E}_{\mathcal{D}}\left[ \mathrm{ALG}_v - \hat{V}_n \right] = \sum_{i=1}^{n} \mathbb{E}_{\mathcal{D}}\left[ (\hat{V}_i - \hat{V}_n) \cdot \mathbb{1}\{\text{stop at } i\} \right]$$

$$= \sum_{i=1}^{n-1} \mathbb{E}_{\mathcal{D}}\left[ (\hat{V}_i - \hat{V}_n) \cdot \mathbb{1}\left\{ \overline{V}^i_{\boldsymbol{\tau}} \wedge \left( \hat{V}_i \leq \tau_{i+1} \right) \right\} \right]$$

$$= \sum_{i=1}^{n-1} \mathbb{E}_{\mathcal{D}}\left[ \tau_{i+1} \mathbf{Pr}_{r \sim U[0,\tau_{i+1}]}\left[ r < \hat{V}_i - \hat{V}_n \right] \cdot \mathbb{1}\left\{ \overline{V}^i_{\boldsymbol{\tau}} \wedge \left( \hat{V}_i \leq \tau_{i+1} \right) \right\} \right]$$

$$= \sum_{i=1}^{n-1} \tau_{i+1} \mathbf{Pr}_{\mathcal{D}, r \sim U[0,\tau_{i+1}]}\left[ \overline{V}^i_{\boldsymbol{\tau}} \wedge \left( r + \hat{V}_n \leq \hat{V}_i \leq \tau_{i+1} \right) \right]$$

In order to show our result, we use from Blumer et al. [1989] that for a class with VC dimension $d < \infty$ that we can learn it with error at most $\varepsilon$ with probability $1 - \delta$ using $m = \mathrm{poly}(1/\varepsilon, d, \log(1/\delta))$ samples.

Consider the class $\mathcal{F}_{\boldsymbol{\tau}}(\hat{V}, r) = \bigwedge_{j=1}^{i-1}(\hat{V}_j > \tau_{j+1})$. This defines an axis parallel rectangle in $\mathbb{R}^i$, therefore its VC-dimension is $2i$. Using the observation above we have that using $m = \mathrm{poly}(1/\varepsilon, n, \log(1/\delta))$ samples, , with probability at least $1 - \delta$, it holds

$$\left| \mathbf{Pr}_{\mathcal{D}}\left[ \mathcal{F}_{\boldsymbol{\tau}}(\hat{V}, r) \right] - \mathbf{Pr}_{\hat{\mathcal{D}}}\left[ \mathcal{F}_{\boldsymbol{\tau}}(\hat{V}, r) \right] \right| \leq \varepsilon$$

for all $\boldsymbol{\tau} \in \mathbb{R}^n$.

Similarly, the class $\mathcal{C}_{\boldsymbol{\tau}}(\hat{V}, r) = \bigwedge_{j=1}^{i-1}\left(\hat{V}_j > \tau_{j+1}\right) \wedge \left( r + \hat{V}_n \leq \hat{V}_i \leq \tau_{i+1} \right)$ has VC-dimension $O(n)$ since it is an intersection of at most $n$ (sparse) halfspaces. Therefore, the same argument as before applies and for $m = \mathrm{poly}(1/\varepsilon, n, \log(1/\delta))$ samples, we get

$$\left| \mathbf{Pr}_{\mathcal{D}, r \sim U[0,\tau_{i+1}]}\left[ \mathcal{C}_{\boldsymbol{\tau}}(\hat{V}, r) \right] - \mathbf{Pr}_{\hat{\mathcal{D}}, r \sim U[0,\tau_{i+1}]}\left[ \mathcal{C}_{\boldsymbol{\tau}}(\hat{V}, r) \right] \right| \leq \varepsilon$$

for all $\boldsymbol{\tau} \in \mathbb{R}^n$, with probability at least $1 - \delta$.

Putting it all together, the error can still be unbounded if the thresholds $\tau$ are too large. However, since we assume that $\tau_i \leq n/\varepsilon$ for all $i \in [n]$, poly$(n, 1/\varepsilon, \log(1/\delta))$ samples suffice to get $\varepsilon$ error overall, by setting $\varepsilon \leftarrow \frac{\varepsilon^2}{n}$.

While we obtain the result for a fixed permutation, we can directly obtain the result for all $n!$ permutations through a union bound. Setting $\delta \leftarrow \frac{\delta}{n!}$ only introduces an additional factor of $\log(n!) = n \log n$ in the overall sample complexity. $\qquad \square$

**Lemma 4.1.2.** *Let $\mathcal{D}$ be any distribution of values. Let $\varepsilon > 0$ and consider a permutation $\pi$ and thresholds $\boldsymbol{\tau}$. Moreover, let $\tau'$ be the thresholds capped to $n/\varepsilon$, i.e. setting $\tau'_b = \min\{\tau_b, n/\varepsilon\}$ for all boxes $b$. Then,*

$$\mathbb{E}_{v \sim D}\left[ALG(\pi, \tau')\right] \leq (1 + \varepsilon)\mathbb{E}_{v \sim D}\left[ALG(\pi, \tau)\right].$$

*Proof of Lemma 4.1.2.* We compare the expected cost of ALG with the original thresholds and the transformed one ALG$'$ with the capped thresholds. For any value vector $\boldsymbol{v} \sim \mathcal{D}$, either (1) the algorithms stopped at the same point having the same opening cost and value, or (2) ALG stopped earlier at a threshold $\tau > n/\varepsilon$, while ALG$'$ continued. In the latter case, the value $v$ that ALG gets is greater than $n/\varepsilon$, while the value $v'$ that ALG$'$ gets is smaller, $v' \leq v$. For such a scenario, the opening cost $c$ of ALG, and the opening cost $c'$ of ALG$'$ satisfy $c' \leq c + n$. Thus, the total cost is $c' + v' \leq c + v + n \leq (1 + \varepsilon)(c + v)$ Overall, we get that

$$\mathbb{E}_{\mathcal{D}}\left[ALG'\right] \leq \mathbb{E}_{\mathcal{D}}\left[ALG\right](1 + \varepsilon).$$

$\qquad \square$

**Theorem 4.1.** *Consider an instance of Pandora's Box with opening costs equal to 1. For any given parameters $\varepsilon, \delta > 0$, using $m = poly(n, 1/\varepsilon, \log(1/\delta))$ samples from $\mathcal{D}$, Algorithm 1 (Variant 1) obtains a $4.428 + \varepsilon$ approximation policy against the partially-adaptive optimal, with probability at least $1 - \delta$.*

*Proof of Theorem 4.1.* With poly$(n, \varepsilon, \log(1/\delta))$ samples from $\mathcal{D}$, we obtain an empirical distribution $\hat{\mathcal{D}}$.

From Lemma 4.1.1, we have that with probability at least $1 - \delta\varepsilon/\log(1/\delta)$, the following holds

$$\left| \mathbb{E}_{v \sim \hat{D}}\left[ALG(\pi, \tau) - \min_{b \in \mathcal{B}} v_b\right] - \mathbb{E}_{v \sim D}\left[ALG(\pi, \tau) - \min_{b \in \mathcal{B}} v_b\right] \right| \leq \varepsilon \qquad (10)$$

for any permutation $\pi$ and any vector of thresholds $\boldsymbol{v} \in \left[0, \frac{n}{\varepsilon}\right]^n$. This gives us that we can estimate the cost of a threshold policy accurately.

To compare with the set of all partially adaptive policies that may not take the form of a threshold policy, we consider the set of scenario aware policies (SA). These are policies SA$(\pi)$ parameterized by a permutation $\pi$ of boxes and are forced to visit the boxes in that order. However, they are aware of all values in the boxes in advance and know precisely when to stop. These are unrealistic policies introduced in Chawla et al. [2020] which serve as an upper bound to the set of all partially adaptive policies.

As shown in Chawla et al. [2020] (Lemma 3.3), scenario-aware policies are also learnable from samples. With probability at least $1 - \delta\varepsilon/\log(1/\delta)$, it holds that for any permutation $\pi$

$$\left| \mathbb{E}_{v \sim \hat{D}}\left[SA(\pi) - \min_{b \in \mathcal{B}} v_b\right] - \mathbb{E}_{v \sim D}\left[SA(\pi) - \min_{b \in \mathcal{B}} v_b\right] \right| \leq \varepsilon. \qquad (11)$$

The $\alpha$-approximation guarantees (with $a \approx 4.428$) of Algorithm 1 hold even against scenario aware policies as there is no restriction on how the partially-adaptive policy may choose to stop. So for the empirical distribution, we can compute a permutation $\hat{\pi}$ and thresholds $\hat{\tau}$ such that:

$$\mathbb{E}_{\hat{D}}\left[ALG(\hat{\pi}, \hat{\tau})\right] \leq \alpha \cdot \min_{\pi} \mathbb{E}_{\hat{D}}\left[SA(\pi)\right]$$

Clipping the thresholds to obtain $\hat{\tau}' = \min\{\hat{\tau}, n/\varepsilon\}$, and letting $\Delta = \mathbb{E}_{v \sim \hat{D}}\left[\min_{b \in \mathcal{B}} v_b\right] - \mathbb{E}_{v \sim D}\left[\min_{b \in \mathcal{B}} v_b\right]$, we have that:

$$\mathbb{E}_D\left[ALG(\hat{\pi}, \hat{\tau}')\right] \leq \mathbb{E}_{\hat{D}}\left[ALG(\hat{\pi}, \hat{\tau}')\right] - \Delta + \varepsilon$$

$$\leq (1+\varepsilon)\mathbb{E}_{\hat{D}}\left[\text{ALG}(\hat{\pi},\hat{\tau})\right] + \Delta + \varepsilon/4$$
$$\leq (1+\varepsilon)\alpha \cdot \min_{\pi} \mathbb{E}_{\hat{D}}\left[SA(\pi)\right] - \Delta + \varepsilon/4$$
$$\leq (1+\varepsilon)\alpha \cdot \min_{\pi} \mathbb{E}_{D}\left[SA(\pi)\right] + O(\Delta + \varepsilon)$$

By Markov's inequality, we have that $\mathbf{Pr}\left[\mathbb{E}_{v\sim\hat{D}}\left[\min_{b\in\mathcal{B}} v_b\right] \leq (1+\varepsilon)\mathbb{E}_{v\sim D}\left[\min_{b\in\mathcal{B}} v_b\right]\right] \geq \frac{\varepsilon}{1+\varepsilon} \geq \varepsilon/2$.

Thus, repeating the sampling process $\frac{O(\log 1/\delta)}{\varepsilon}$ times and picking the empirical distribution with minimum $\mathbb{E}_{v\sim\hat{D}}\left[\min_{b\in\mathcal{B}} v_b\right]$ satisfies $\Delta \leq \varepsilon\mathbb{E}_{v\sim D}\left[\min_{b\in\mathcal{B}} v_b\right]$ with probability at least $1-\delta$ and simultaneously satisfies equations (10) and (11).

This shows that $\mathbb{E}_{D}\left[\text{ALG}(\hat{\pi},\hat{\tau}')\right] \leq (1+O(\varepsilon))\alpha \cdot \min_{\pi} \mathbb{E}_{D}\left[SA(\pi)\right]$ which completes the proof by rescaling $\varepsilon$ by a constant.

$\square$

