# OpenReview forum: "Weitzman's Rule for Pandora's Box with Correlations"
_NeurIPS.cc/2023/Conference — NeurIPS 2023 poster_

### Official Review · Reviewer_PoqM · 2023-06-27

**Soundness:** 4 excellent
**Presentation:** 3 good
**Contribution:** 4 excellent
**Rating:** 8
**Confidence:** 4

**Summary:**

This paper considers Pandora’s Box problem with correlated values. Previous work gives a 9.22 approximation for this problem. This problem considers two variants, depending on whether the algorithm updates based on exact values or on the event that the value is large, with approximation factors of 5.828 and 4.428, respectively. The latter approach can be extended to the case of unknown distributions that the algorithm has sample access to. An interesting feature of the main algorithmic blueprint is that it is a direct extension of Weitzman’s original rule for this problem (in the independent value case).

**Strengths:**

This paper makes a major contribution to optimal stopping theory, by giving a clean answer to a very natural and important problem.

**Weaknesses:**

The technical writing can definitely be improved. For example, it is very difficult to follow the histogram argument in the proof of Theorem 3.2 without figures.

(
Some typos/writing notes since there is nowhere else to put them:
- Typo in line 41: “are are”
- I believe the notation (x)^+ is not defined anywhere.
- It’d be nice to do some of the math slower. E.g., in 154, it’d be useful for the reader to write ALG first as an expectation, and then slowly open it up and use the uniform fact, etc
- Typo in line 274: the \geq sign should not be there
)


**Questions:**

Why are both variants considered? It seems like variant 1 is better under every aspect considered (approximation ratio, simpler proof, learnable).

---

> ### Author Rebuttal · Authors · 2023-08-09
>
> We thank the reviewer for the comments and typos caught; we will add a figure in the final version showing how the histogram method works, to better illustrate the proof.
>
> To answer the question: when we initially started considering variant 2, we were not sure whether it would give a better approximation or not. It is however a more natural variant: we don’t keep scenarios that are clearly not possible anymore (which variant 1 does).
>
> As other reviewers pointed out, intuitively we would expect a better factor. However since the algorithm is a greedy approximation, we soon realized that the factor may not necessarily be monotone on the amount of information given. Variant 2, led us to a generalization of the histogram proof on trees, which might be of independent interest, and therefore we decided to include it as well. Additionally our analysis is not necessarily tight, and a better factor might still be possible.

---

> > ### Comment · Reviewer_PoqM · 2023-08-15
> >
> > Thank you for your reply.

---

### Official Review · Reviewer_1WDf · 2023-06-30

**Soundness:** 2 fair
**Presentation:** 2 fair
**Contribution:** 2 fair
**Rating:** 4
**Confidence:** 3

**Summary:**

This paper considers the Pandora's box problem with correlated values. In each step, the algorithm chooses an unopened box and observes its value generated from the known distribution. The goal is minimizing the sum of the minimum value among the opened boxes and the total opening cost. The distribution is given as a uniform distribution over finite possible scenarios. This paper shows simple greedy algorithms based on reservation values achieve improved approximation ratios. The existing algorithms for this problem are based on more complicated techniques. This paper also proposes an algorithm that uses samples from the distribution instead of its explicit representation.


**Strengths:**

The Pandora's box problem is an interesting problem that has been extensively studied in TCS and algorithmic game theory. Its correlated version is regarded as a noteworthy technical challenge. This paper improves and simplifies the existing results. This is a significant theoretical advancement.


**Weaknesses:**

I have a concern regarding the soundness of the proofs. The proofs sometimes appear informal, making it challenging to verify their validity. For example, the description of the algorithm is not always clear. In Algorithm 2, the opening cost of the already opened boxes are set to 0 (line 8). Then their reservation value might become the minimum, which is selected in line 4-5. The algorithmic behavior in this case is not clearly specified. This issue is relevant to the analysis around the inequalities (4) and (5). The opening cost $c_b$ appears in these inequalities, but it is not constant during the execution of the algorithm.



**Questions:**

What is the formal treatment of the opening cost $c_b$ in the analysis?


**Limitations:**

Yes, it is adequate.

---

> ### Author Rebuttal · Authors · 2023-08-09
>
> We thank the reviewer for the comments, and we will clarify that a box can be opened many times during the run of the algorithm.
>
> * Regarding the algorithm’s behavior: a box can be opened many times, therefore after we open it once (and make the cost 0) it could be re-opened at a later stage of the algorithm. This is similar to what happens in the independent case algorithm (Weit79) where we could stop at any point and “go back” to pick a value seen in a box opened before. This is the same behavior: we can go back and select a box previously opened with no extra cost.
>
> * The fact that the opening cost becomes 0 is not directly used in the analysis (i.e. inequalities (4) and (5) ). This means that our analysis gives an upper bound on the cost of the algorithm, *even if* the algorithm *never* changes the cost of an opened box to 0. That is the reason in inequalities (4) and (5) the cost appears unchanged but the analysis still works for the algorithm since we just want an upper bound (and if we changed the cost to 0 this would only lower the cost of the algorithm).
>
> Therefore, our analysis shows that we can bound the algorithm’s cost even without setting costs to 0. However, setting them to 0 this makes it easier to see how it directly generalizes the independent case (since it mimics the “going back to pick a box opened earlier” behavior).

---

> > ### Comment · Reviewer_1WDf · 2023-08-17
> > **Comment**
> >
> > Thank you for addressing my concerns in your response. The clarification provided regarding the opening cost is appreciated. To enhance the clarity of this aspect, I recommend considering a revision of the pseudocode and the proofs. Given that there is room for improvement in the writing, my score is still on the borderline.

---

> > > ### Author Response · Authors · 2023-08-21
> > > **Thanks for comment**
> > >
> > > Thank you for reading our reply. We would like to mention that the clarification of the opening's cost role does not require a major rewrite and can easily be incorporated in the final version of the paper.
> > >
> > > Specifically, as we described in our reply, the cost change from $c_i$ to 0 does not come into the proof at all, therefore the technical part/proofs do not need any major changes. Of course we acknowledge that it is unclear why it is of no consequence to the proof, and what is its role in the algorithm. We intend to add a paragraph discussing
> > > * why it does not appear in the proof (so that someone reading the proof will understand why the change to 0 does not appear),
> > > * clarify its role in the algorithm, by also mildly editing the pseudocode (So that someone reading the pseudocode has no doubt on whether a box can be reopened),
> > >
> > > which is an easily implementable change to do for the final version.

---

### Official Review · Reviewer_coaw · 2023-07-06

**Soundness:** 2 fair
**Presentation:** 2 fair
**Contribution:** 3 good
**Rating:** 6
**Confidence:** 2

**Summary:**

This paper provides an exploration of Pandora's box problem with correlations. The authors innovatively modify the computation of reservation values within Weitzman's algorithm. They further solidify their contribution by proving the approximation ratio of the proposed algorithms under various distribution updating schemes.

**Strengths:**

This paper provides an exploration of Pandora's box problem with correlations. The authors innovatively modify the computation of reservation values within Weitzman's algorithm. They further solidify their contribution by proving the approximation ratio of the proposed algorithms under various distribution updating schemes. The algorithms presented are intriguing and succinct, which I find very appealing.
The whole paper also reads very well except skipping of some technical parts.

**Weaknesses:**

However, I admit that some aspects of the algorithms and their details elude my understanding. I would greatly appreciate further explanations to deepen my comprehension. Specifically, it remains unclear what it means to update the value distribution $\mathcal{D}$ conditional on $V_b > \sigma_b$ and $V_b = \sigma_b$. Why even we could have both different updating? I am also uncertain how this conditional updating of $\mathcal{D}$ gives rise to Algorithms 2 and 3.

Providing additional insights into these elements would undoubtedly clarify the understanding for readers like myself and further enhance the value of your work.

**Questions:**

I would greatly appreciate further explanations to deepen my comprehension. Specifically, it remains unclear what it means to update the value distribution $\mathcal{D}$ conditional on $V_b > \sigma_b$ and $V_b = \sigma_b$. Why even we could have both different updating? I am also uncertain how this conditional updating of $\mathcal{D}$ gives rise to Algorithms 2 and 3.

Providing additional insights into these elements would undoubtedly clarify the understanding for readers like myself and further enhance the value of your work.

---

> ### Author Rebuttal · Authors · 2023-08-09
>
> We thank the reviewer for the comments, we can add the appropriate clarifications in the final version. We answer the question below.
>
>  The (correlated) distribution is a set of vectors of size $n$ (described end of page 3), where each is drawn with some probability. When we open a box and see a value, some scenarios are not “possible” anymore, i.e. we know they cannot be the ones realized. We illustrate in the following example. Assume there are 3 of these vectors (scenarios).
>
> |    | B1 | B2 | B3 |
> |:--:|:--:|:--:|----|
> | **S1** |  3 |  4 | 7  |
> | **S2** |  6 |  4 | 2  |
> | **S3** | 7  | 7  | 2  |
>
> The rows in the matrix above are the scenarios, and the columns are the boxes. For example, if scenario S2 is the one realized (i.e. drawn from the distribution) then the values inside boxes B1,B2 and B3 are 6, 4 and 2 respectively. The distribution $D$ is essentially drawing one of the scenarios with some probability.
>
> To see what the conditioning means: assume we open box 1 and we see the value 6 (and assume for the sake of the example that the reservation value of box 1 is $\sigma_1 = 5$).
>
> * Variant 1: we condition on $6=V_b> \sigma_1 = 5$, meaning that scenario S1 is not possible anymore (because if S1 was the one drawn from D, then we would have seen a value less than $\sigma_1 = 5$ when opening the box), and is removed from the set S the algorithm considers (line 9, Alg 2)
>
> * Variant 2: we condition on $V_b = 6$, which means that scenarios 1 and 3 are both removed (similarly, because if any of these were drawn, we would not have seen 6 upon opening the box)
>
> The second way these variants differ, is that due to this conditioning, the solution for the $V_b> s$ variant is “partially adaptive” meaning that the next box the algorithm opens, *only* depends on the *scenarios that remain*. However, for the $V_b=v$ variant the solution is “fully adaptive” (meaning that the next box opened, *depends on the exact value seen*). This is illustrated in Figures 2 and 4 of the appendix (supplementary material), where variant’s 1 solution can be represented by a line graph (Fig 2), while Variant’s 2 solution is a tree (Fig 4).

---

> > ### Comment · Reviewer_coaw · 2023-08-20
> >
> > The provided example is highly illuminating and resonates well with me. I would strongly recommend incorporating this example, along with the accompanying reasoning, into the final version of the paper. Doing so would undoubtedly aid in enhancing the readers' comprehension. While I acknowledge that the authors have made substantial efforts in addressing queries, I concur with my fellow reviewers that the paper still requires structural improvements for better readability. Therefore, I am inclined to maintain my current review score.

---

### Official Review · Reviewer_ktwT · 2023-07-07

**Soundness:** 3 good
**Presentation:** 2 fair
**Contribution:** 3 good
**Rating:** 6
**Confidence:** 4

**Summary:**

This paper studies the problem of Pandora's boxes with the values of the boxes correlated. The authors extend the classical algorithm Weitzman’s Rule for independent values of boxes to the correlated case, and propose new algorithms with better approximation than previous works that are learnable from samples.

**Strengths:**

Novelty: The major novelty of this paper is to extend the Weitzman’s Rule algorithm to the cases of correlated boxes' values, and simplify the problems greatly. Though none of the correlated case or the algorithm are new, this extension does have its unique value by making simplicity and improving the approximation guarantee.

Quality: This paper's result is sound and the analysis looks good. The results are strong as well.

**Weaknesses:**

Clarity: This paper is not very clear in some places. For instance, it does not give the definition of \epsilon-approximation before first using it, which may make some readers hard to understand this paper. The proofs are also too simplified without enough intermediate steps to make readers easy to follow.

Significance. This is a minor weakness, where the authors need to better justify why the correlated case is important. What is the major motivation in practice, or is is a theoretical fundamental problem that can solve a series of dependent problems?

**Questions:**

Justify the major motivation of the correlated case for its practical or theoretical importance.


**Limitations:**

As written in Weakness

---

> ### Author Rebuttal · Authors · 2023-08-09
>
> We thank the reviewer for the comments, and we will clarify and define the notions that aren’t currently clear. To answer the question: on the theoretical side, removing the strong assumption of independence generalizes the problem, and gave rise to new techniques that may be used to solve similar problems with correlation (e.g. prophet inequalities [1]).
> On the practical side, in real life  independence is an assumption that usually does not hold. For example:
> 1. *Housing market*: we are looking to buy a house and we have a list of potential properties in mind, and have to decide on some order to visit them, learn more information and finally buy one. The houses are the boxes, we have to spend time/gas driving to each property (i.e. pay the opening cost) in order to learn the actual price (i.e. the house might need more repairs, therefore the price is not as low as we thought). The houses’ prices however are not independent:
>     1. Location affects nearby houses the same way (e.g. if a neighborhood is bad, all houses near there will be affected the same way)
>     2. Having the same property manager (e.g. a potentially dishonest property manager hides the issues with the house, meaning that houses with the same property manager are potentially affected the same way)
>
> 2. *Acquiring an item*: we want to buy an item that many different shops sell. We have a prior belief for every shop (e.g. if it’s generally expensive or not) but we have to spend time inquiring about the price. However, some shops will use the same supplier, meaning that if the supplier is low on some part used for the item, these shops will sell it at higher price, compared to the ones using different suppliers.
>
> 3. *Job Search*: we are a company interviewing candidates for a job position. Each candidate is a box, we have some prior for each (e.g. their cv) but until we spend time to interview them (i.e pay the opening cost) we don’t know their exact value (i.e. how close they match the position). Our goal is to select the best candidate without interviewing too many. Correlations arise since some of them have common qualifications in their cv that affect their value (e.g. they have graduated the same university, which happens to offer a program that matches very well the position)
>
> These examples are the motivation that started this area of research in economics (consumer search, housing market, job search), where optimal search problems (like Pandora’s Box) arise [see [2] for more details].
>
> [1] Prophet Inequalities with Linear Correlations and Augmentations [Immorlica, Singla, Waggoner] EC 20
>
> [2] The Economics of Search [Brian McCall, John McCall], Routledge, 2007]

---

### Official Review · Reviewer_6o9F · 2023-07-19

**Soundness:** 3 good
**Presentation:** 3 good
**Contribution:** 3 good
**Rating:** 7
**Confidence:** 4

**Summary:**

This paper studies the Pandora’s box problem with correlation. The problem is as follows: a decision maker is presented with $n$ boxes to explore, and each box $b_i$ is associated with a hidden value $v_i$ and a known cost $c_i$ that needs to be paid to reveal the value. The values in the boxes are drawn from a known correlated distribution. The decision maker opens the box one after the other in an arbitrary order that may depend on the realized values uncovered and, when it decides to stop, pays the total cost of exploration (the sum of the costs of the boxes it opened) plus the smallest uncovered value. The goal of the problem is to design a strategy that minimizes the cost paid by the agent.

The authors present two algorithms and a learnability result. A first algorithm yields a constant factor approximation (4.428) to the best non-adaptive strategy when the only information the decision maker uses to update its strategy is whether the exploration stopped or not, according to some stopping rule. The second algorithm yields another constant factor approximation (5.828) to the same benchmark in a full update model (the decision maker updates its prior distribution according to the exact realized values). Both approximations improve on the state of the art.

**Strengths:**

The Pandora’s Box problem is an exciting and challenging model for exploration under uncertainty that has received much attention in recent years (see, e.g., the recent papers at STOC and EC). The correlated version of the problem is interesting and overcomes the unnatural assumption of independent valuations made in the original model by Weitzman.

Strengths:
- The algorithms proposed by the authors are uncomplicated and enjoy the desirable property of extending Weitzman’s notion of reservation value. This fills a gap in understanding the correlated version of the problem, as previous works used different techniques, achieving worse results.
- The learnability result is interesting and useful in overcoming the natural limitation of not knowing the underlying distribution. The result complements its analogous for the independent scenario (COLT 21, ‘‘Generalizing complex hypotheses on product distributions: Auctions, prophet inequalities, and Pandora’s problem’’).

**Weaknesses:**

- It is unnatural to obtain weaker results (against the same benchmark) when the decision-maker employs a richer update rule. The main weakness of the paper is not providing a convincing explanation of this fact. Moreover, it is impossible to understand from the main body what is the technicality that allows the improved approximation factor.
- Removing the abstract from the manuscript to fit into the 9 pages limit is a borderline practice.


Minor comments:
- Please be consistent in the use of \citet and \citep. Use \citet when the article cited is part of the sentence (e.g., in the abstract), and \citep otherwise.
- Please explain why the second to last display of page 3 implies the last one. Why is it safe to make the sum_s \in A P_D(s) term appear at the numerator without affecting the maximization problem?

**Questions:**

1. Why consider the richer update model? It gives a worse approximation guarantee. The decision-maker should ignore it

2. Is there any lower bound on the approximation factor? How far are the proposed results from the optimal poly-time algorithm?

3. Please explain why the second to last display of page 3 implies the last one. Why is it safe to make the sum_s \in A P_D(s) term appear at the numerator without affecting the maximization problem?

-----------------------

Raised my score after rebuttal

**Limitations:**

See above

---

> ### Author Rebuttal · Authors · 2023-08-09
>
> We thank the reviewer for the comments, we are going to fix the citation inconsistencies, and add more intuition on the difference between the two algorithms versions. Also apologies for the abstract, the author who submitted the paper was careless with copying everything to the NeurIPS template. Space was not an issue, since we could have moved more proofs to the appendix. To answer the reviewer’s questions:
>
> 1. When we initially started considering the richer update model (Variant 2), we were not sure whether it would give a better approximation or not. It is however a more natural variant: we don’t keep scenarios that are clearly not possible anymore (which variant 1 does). Intuitively we would expect a better factor. However since the algorithm is a greedy approximation, we soon realized that the factor may not necessarily be monotone on the amount of information given. The richer update model led us to a generalization of the histogram proof on trees, which might be of independent interest, and therefore we decided to include it as well. Additionally our analysis is not necessarily tight, and a better factor might be possible for the richer updates variant (even though even the worse approximation for the richer update variant is close to the lower bound --> see below)
>
> 2. The best possible for a polynomial-time algorithm is a $4$-approximation, and this is a result from the Min Sum Set cover problem [FLT04], which means our algorithm is almost tight. We have already included a discussion on the lower bound in the Appendix (supplementary material, section A.3). We will however mention it in the main text too, for completeness.
>
> 3. The maximization problem is not affected because we need the max over subsets $A\subseteq S$ such that the expression is equal to $0$; i.e. the expression is the following:
> { Find $\sigma $ that is the solution to $\max_{A\subseteq S} { f(A,\sigma) } = 0$} where $f(A,\sigma)= \sum_{s\in A} Pr_D[s] (\sigma_b-v_b^s)-c_b$
> Therefore, by dividing with something positive (i.e. $\sum_{s \in A} Pr_{\mathcal{D}}[s]$), we still need the numerator to be $0$ for this to be satisfied, which is not affected by dividing by any positive number. Note also that $S$ is not empty (since if it was, the algorithm would have stopped), so this is a well defined division.

---

> > ### Comment · Reviewer_6o9F · 2023-08-21
> >
> > I thank the authors for their rebuttal, and I raise my score accordingly.

---

### Decision · Program_Chairs · 2023-09-21

**Decision:**

Accept (poster)

**Comment:**

This paper considers a well-motivated extension of the Pandora's box problem that allows for correlations. The reviewers agree that the setting considered is natural, that the results are strong, and that the techniques are interesting. I recommend that this paper be accepted.

Multiple reviewers found that the clarity of certain parts of the paper could be improved, please address these comments in the camera-ready version of the paper.